# Quality of Beef Burgers Formulated with Fat Substitute in a Form of Freeze-Dried Hydrogel Enriched with Açai Oil

**DOI:** 10.3390/molecules27123700

**Published:** 2022-06-09

**Authors:** Monika Hanula, Arkadiusz Szpicer, Elżbieta Górska-Horczyczak, Gohar Khachatryan, Ewelina Pogorzelska-Nowicka, Andrzej Poltorak

**Affiliations:** 1Department of Technique and Food Development, Institute of Human Nutrition Sciences, Warsaw University of Life Sciences, Nowoursynowska 159c Street 32, 02-776 Warsaw, Poland; arkadiusz_szpicer@sggw.edu.pl (A.S.); elzbieta_gorska_horczyczak@sggw.edu.pl (E.G.-H.); ewelina_pogorzelska@sggw.pl (E.P.-N.); andrzej_poltorak@sggw.edu.pl (A.P.); 2Department of Food Analysis and Evaluation of Food Quality, Faculty of Food Technology, University of Agriculture in Krakow, Mickiewicz Ave. 21, 31-120 Krakow, Poland; gohar.khachatryan@urk.edu.pl

**Keywords:** fatty acids profile, substitute fat, açai oil, health index, oxidation

## Abstract

The growing number of people at high risk of cardiovascular disease development contributed to both changes in diets by consumers and the reformulation of food products by food producers. Cardiovascular diseases are caused by the i.a. consumption of meat that contains animal fat rich in saturated fatty acids (SFA). The use of fat substitutes in meat seems to be a promising tool for the reduction of cardiovascular disease occurrence. In the presented study, beef fat was replaced at 0 (CO), 25 (S-25%), 50 (S-50%), 75 (S-75%), and 100% (S-100%) by a fat substitute in a form of a lyophilized hydrogel emulsion enriched with encapsulated açai oil. The chemical (TBARS, volatile compound profile, fatty acid profile, pH), and physical (TPA, consumer rating, L*a*b* color, cooking loss) analyses were performed on raw and grilled burgers subjected to storage at cold conditions (4 °C) in days 0 and 7. Burgers formulated with hydrogels had a higher content of polyunsaturated fatty acids (PUFAs) of about 32% (*p* < 0.05) and reduced SFAs by 22%. Reformulation of the burger resulted in lower nutritional indices of the atherogenicity index (AI) (0.8 for CO, 0.3 for S-100%, *p* < 0.05) and thrombogenicity index (TI) (1.8 for CO, 0.6 for S-100%, *p* < 0.05), as well as led to an increased h/H ratio (1.3 for CO, 3.9 for S-100%, *p* < 0.05). Furthermore the application of freeze-dried hydrogels reduced cooking loss. Moreover, consumers did not observe significant differences (*p* < 0.05) between the control and S-25% and S-50% burgers. Thus, the use of lyophilized hydrogels formulated with konjac flour and sodium alginate and enriched with encapsulated acai oil can be successfully applied as a fat substitute in beef burgers.

## 1. Introduction

Burgers have become very popular around the world thanks to their quick preparation and high sensory acceptability. However, these products typically have a high animal fat content (20−30%). Animal fat is rich in saturated fatty acids (SFA). According to the World Health Organization, the very high intake of saturated fats has become a serious problem because SFA contributes to an increased risk of cardiovascular disease. For this reason, it is recommended to limit the intake of SFA [1,2,3]. Therefore, to ensure that meat products are more nutritionally beneficial, many studies have been conducted to reformulate meat products [4,5,6,7,8]. Ingredients have been changed by seeking alternative solutions to improve the lipid profile. One strategy to improve fatty acid profile values is to reduce SFA by limiting animal fat and/or using vegetable oils that enrich the product with MUFA and PUFA [2].

In this context, the use of fat substitutes is a promising method; however, the challenge is to use a substitute that does not negatively affect the consumer acceptance of the product while improving the health quality of the meat product.

Vegetable oil can be immobilized or encapsulated in hydrogels or oleogels [9]. Oleogels have been the subject of many studies, but due to their high production costs and the destructive effect of organogelators on the fatty acid profile (high polymerization temperature), these substances are very difficult to use in the food industry. In contrast, the use of emulsion hydrogels is cheaper when compared to oleogels and does not require high temperatures. Therefore, hydrogel emulsions are used for the immobilization/encapsulation of thermally sensitive compounds. Moreover, the oil content in hydrogel emulsions is <50%. By using this type of fat substitute, the fatty acid profile can be improved and the total fat content reduced. Encapsulation is an effective solution to increase the chemical and biological stability of active substances and protect them from unavoidable reactions in food systems. However, hydrogel, due to its high water content, has a limited lifetime related to the stability of the hydrogel system. Lyophilization of the hydrogel emulsion seems to be promising, as it extends the shelf life of the hydrogel emulsion without negatively affecting the bioactive compounds [9,10].

Increased consumer awareness of the impact of food on health, disease incidence, and overall well-being has contributed to the rapid development of functional foods and has increased interest in ‘superfood’ products. Recently, açai berries that are widely used in food products have enjoyed great popularity [11]. The extracted oil of the açai berry, despite its high PUFA content [12], is characterized by antiproliferative and anti-inflammatory properties and it has no genotoxic effects [13]. This oil also contains polyphenolic compounds, such as vanillic acid, ferulic acid, catechin, and syringic acid [14,15].

Through analyzing the above data, the food industry is currently facing difficulties in reducing the fat content of burgers. A reduced fat content in meat products can have a number of negative effects on sensory and process quality, such as increased cooking losses, reduced yield, altered texture, and reduced juiciness and flavor. In previous studies, fat substitutes in the form of hydrogel emulsions with encapsulated oil by coacervation was shown to improve the physical and chemical properties of beef burgers (Hanule et al. 2022). However, taking into account the fact that fat substitution in the form of hydrogel emulsions may not necessarily be a convenient form for their storage, transport, and introduction into the final product from a technological point of view, it was decided to look for a method to change the consistency of fat substitutes without having a destructive influence on the encapsulated oil. Therefore, the purpose of this study was to test the effect of freeze-dried fat substitute with encapsulated açai oil, on the quality of formulated beef burgers.

The aim of this study was to develop a reformulated burger with a reduced SFA content by using a fat substitute in the form of a freeze-dried hydrogel emulsion with encapsulated açai oil.

## 2. Results and Discussion

### 2.1. The Analysis of the Freeze-Dried Hydrogel Emulsion Using SEM

The SEM analysis of the freeze-dried hydrogel emulsion is shown in Figure 1. The SEM analysis showed that nanocapsules were obtained in a polysaccharide matrix and the freeze-drying process did not negatively affect their structure. The process of encapsulating açai oil in polymers (konjac flour and sodium alginate) resulted in the formation of multilayer shells. Multilayered nanocapsules were found throughout the matrix and their distribution was uniform.

### 2.2. Physical Analyses of the Beef Burger

#### 2.2.1. Color and pH Analysis

The first parameter that characterizes meat quality and that consumers pay attention to when choosing meat is its color. The results of the color analysis of raw and grilled burgers are shown in Table 1. In the analysis of the color parameters of raw burgers on day 0, the reformulations had no effect on the color parameters L* and a*, while the value of the parameter b* increased with the increase in the percentage of a fat substitute. After 7 days of storage, the brightness parameter (L*) was highest for CO. With the increasing amount of tallow replacement, a slight decrease in the brightness parameter was observed. The fat replacement used, which was freeze-dried hydrogel, probably contributed to moisture binding, while the control sample probably had a higher drip loss, which increased the brightness of the burger by affecting its chromaticity [16,17,18]. Moreover, the parameter a* decreased significantly with storage time for both CO and fat-replacement burgers (*p* < 0.05). However, the smallest difference between day 7 and day 0 was observed in the S-100% variant. Furthermore, the parameter a* in raw burgers on day 7 increased with an increasing percentage of fat replacement. The value of parameter b* decreased with storage time in all reformulations used. Nevertheless, the fat replacement contributed to an increase in the value of parameter b* when compared to CO. The decreases in the values of the color parameters a* and b* during storage indicate that the loss of redness is due to the conversion of oxymyoglobin to methemoglobin by free radicals. Free radicals are formed during lipid oxidation and interact with the heme group of myoglobin, initiating the oxidation of the molecule and causing the product to lose its color. Thus, free radicals formed during oxidation can damage the muscle fibers and reduce the pigmentation of the meat [17,19]. The results obtained suggest that the red color in the reformulations used is not only due to changes in myoglobin, but is also an effect of the fat substitute (Table 1). However, when analysing the differences between grilled burgers, the use of fat substitute contributed to higher values of a* (8.47–8.89 for S-100%) and b* (8.33–10.86 for S-100%). When comparing the results of the color parameters L*a*b* of grilled burgers with those of raw burgers, all reformulations showed lower values of the analyzed parameters after grilling. A similar trend was found in the study by Wang et al. [20]. This result is probably due to the heat treatment during which caramelization, protein denaturation and the Maillard reaction take place [21,22]. The values of the parameter ΔE (data not shown) for the variants during storage for raw burgers were 9.3 for CO, 5.26 for S-25%, 3.9 for S-50%, 3.3 for S-75%, and 3.8 for S-100%. According to Altmann et al. [23], which was a detailed literature analysis of ΔE in raw meat, it was concluded that a color difference as low as one is distinguishable by consumers. The results obtained indicate that there are significant differences in color change with storage time. In raw burgers, the variants containing ≥50% fat replacement had the least difference. This difference may be due to the form of the added vegetable oil, which was encapsulated. Moreover, these capsules were cross-linked in the vegetable polymer, causing limited access to factors accelerating fat oxidation processes (light, oxygen).

The effects of the reformulations on the pH of raw burgers at 0 and 7 days of storage are shown in Table 1. The analysis shows that the pH increased with increasing storage time regardless of the type of reformulation used. A similar result was found in the study by Sharaf et al. [24] and Soares et al. [25]. The increase in pH during storage is related to protein degradation and to the formation of volatile compounds of the nitrogen, amine or hydrogen sulfide group [26]. Nevertheless, the fat replacement in the form of a freeze-dried hydrogel emulsion with encapsulated oil contributed to an increase in pH when compared to the control on day 0 (5.38 CO, 5.50–5.56 S-25%-S-100%). This effect is probably related to the pH of the fat replacement used. On the other hand, the slowing down of the pH increase during storage could be due to water retention in the spaces of the freeze-dried biopolymer.

#### 2.2.2. Texture Analysis and Cooking Loss

The fat replacement changed the texture parameters; the results are shown in Table 2. The analysis of parameters such as hardness, springiness, and cohesiveness showed significant differences between the different reformulations (*p* < 0.05). The control variant (CO) had the highest texture parameters (hardness, springiness, cohesiveness) after both 0 and 7 days of storage (102.36–75.82 (N), 0.53 (-), 0.38 (-), respectively). With the increasing substitution of beef tallow by the fat replacement, the studied parameters decrease significantly (*p* < 0.05). From the data obtained, it follows that the tallow content in the burger increases the values of the texture parameters. Similar results were found in the studies by Moghtadaei et al. [27], Zetzl et al. [28], and Paglarini et al. [29]. On the other hand, literature analysis shows an ambiguous trend. The studies carried out by Foggiaro et al. [30], Alejandre et al. [31], and Heck et al. [2] demonstrate an opposite trend. Studies have shown that the main factor influencing the texture of food products is the fat structure. The physicochemical properties of fat and its interaction with meat may explain the differences between the reformulations used and the differences resulting from the type of fat substitute used [1].

Figure 2 shows the effect of fat replacement in the form of a freeze-dried hydrogel emulsion with encapsulated oil on cooking loss parameters. Cooking losses are the main problem when replacing animal fat with oils in meat products [32]. The highest losses when grilling beef burgers occurred in CO after 0 days and 7 days of storage (34 and 31%, respectively). Moreover, this parameter decreased significantly (0 storage day, *p* < 0.05) depending on the degree of animal fat replacement (S-25%, S-50%, S-75%, S-100%). This trend is confirmed by studies conducted by Moghtadaei et al. [26], who used ethylcellulose oleogel as a fat replacement. The decrease in cooking losses during storage observed for all reformulations used is probably related to the loss of intrinsic water during storage as well as the water-binding property of the freeze-dried hydrogel emulsion [27,33].

### 2.3. Chemical Analyses of the Beef Burger

#### 2.3.1. Oxidation Lipids Analysis

The results of the lipid oxidation analysis of burgers with fat substitutes are presented in Figure 3. The values obtained for TBARS analysis ranged from 0.162 (CO, day 0) to 0.429 mg MDA/kg sample (S-75%, day 7). The key factors of lipid oxidation in meat products are the fat content and the fatty acid profile [34]. Therefore, the resulting differences in TBARS values (day 0) in reformulated burgers could be due to the different proportions of fat used and thus the varying fatty acid profile. In our study, the degree of fat oxidation after 7 days of storage was higher in reformulations with 25, 50, and 75% beef fat replacement (S-25%, S-50%, S-75%). This result is probably related to the different susceptibility of fatty acids to the oxidative process [34]. This result is confirmed by the studies of Lorenzo et al. [35], Fonseca et al. [36], and Hautrive et al. [37], in which an increase in fat oxidation was observed in meat products with an increased content of unsaturated fatty acids when compared to saturated fatty acids. The results obtained (S-100%) show that structuring the added fat in the form of a freeze-dried hydrogel emulsion with encapsulated açai oil helped to slow down the oxidation process [31]. The lipid oxidation process leads to a reduction in the specificity and sensory quality of meat products. This is due to an increase in the spectrum of aromas and off-tastes (rancidity) as well as a decrease in the color parameters and texture, which affect the consumer acceptance of the product [34,38]. The threshold value that dictates the loss of sensory quality of food is >1.0 mg MDA/kg burger. The study showed that the TBARS values increased with storage time, regardless of the burger reformulation used. Nevertheless, the values of lipid oxidation after 7 days of storage (0.279–0.429 mg MDA/k sample) were significantly below the set threshold value.

#### 2.3.2. E-Nose Analysis

A statistical quality control plot of the differences in the volatile compound profiles of the reformulated burgers after 0 and 7 days of storage is shown in Figure 4. The analysis of the raw burger after 0 and 7 days of storage shows that significant changes in the volatile compound profile occur with increasing storage time. The analysis shows that the control variant (CO) had the lowest number of aroma units on day zero when compared to the other burger reformulations used. Replacing tallow with a fat substitute contributed to an increase in the number of aroma units, with the highest number being in the S-50% variant. However, when comparing the number of aroma units in CO and S-100% variants in raw burgers, it was found that the transformations due to storage time produced more volatile compounds in tallow than the fat substitute used. When analyzing the volatile compound profile in burgers grilled on day 0, the CO and S-25% reformulations had similar numbers of aroma units. Replacing fat with ≥50% with a freeze-dried hydrogel emulsion with encapsulated açai oil significantly increased the number of aroma units in burgers grilled on day 0. The analysis of the changes in grilled burgers after 7 days of storage indicates that the CO aroma profile was significantly different from S-25%, S-50%, S-75%, and S-100%. Furthermore, the heat treatment significantly reduced the distance of the aroma units in grilled burgers when compared to raw burgers. Moreover, when analyzing the grilled burgers, the opposite trend to raw burgers was observed, i.e., as the replacement of tallow with freeze-dried hydrogel increased, the distance of the aroma units increased. The main groups of compounds formed during lipid oxidation are aldehydes, ketones alcohols, esters, and acids. These are formed when unsaturated fatty acids react with molecular oxygen, resulting in hydroperoxides that are unstable and decomposable [39,40].

Figure 5 shows the relative peak area (%) of the main groups of volatile compounds, i.e., aldehydes (3C), ketones (3A), and alcohols (3B) in raw and grilled burgers with reformulations at day 0 and after 7 days of storage. The data presented show that the fat substitute used (S-100%) generates more volatile compounds belonging to the ketone group than CO for both the raw and grilled burgers on days 0 and 7 of storage. Nevertheless, in raw burgers, the variant S-50% had the highest number of ketone compounds. For grilled burgers, the number of ketone compounds remained at a similar level regardless of the storage time (S-75%, S-100%, *p* < 0.05). The content of volatile compounds belonging to the alcohol group was significantly lower in grilled burgers than in raw burgers. In raw burgers, the lowest number of compounds belonging to the alcohol group was observed for the S-50% variant (day 0 and 7) and the highest was observed for the control variant (day 0). In contrast, for grilled burgers, the number of aldehyde compounds increased significantly on day 7, especially for the control variant (60%) This observation is consistent with the study by Narsaiah et al. [41], which confirms an increased contribution of aldehydes to the volatile compound profile resulting from fat oxidation reactions and thermal processes such as Strecker degradation during grilling.

#### 2.3.3. The Analysis of Fatty Acid Compounds and Health Indicators

The analysis of the profiles of SFA, PUFA, and MUFA in grilled and raw burgers after 0 and 7 days of storage is presented in Figure 6. When a fat substitute in the form of an encapsulated açai oil in freeze-dried hydrogel emulsion was used, the fatty acid profiles were improved. The CO variant had a profile of 52.5% SFA, 44.4% MUFA, and 3.1% PUFA. It was observed that the fatty acid profile changed with an increasing fat substitution (*p* < 0.05). S-100% burgers had the highest PUFA content (25.0% SFA, 34.4% MUFA, 40.6% PUFA). Analysis of the fatty acid profile showed that the PUFA content decreases with storage time. This is due to oxidative processes during storage, where the free radicals oxidize the double bonds of the fatty acids (PUFA) [32,34]. Heat treatment such as grilling significantly affects the fatty acid profile through the occurrence of Maillard reactions. The analysis shows that the fat replacement used contributed to an improvement in the fatty acid profile of the grilled burgers. With increasing fat substitution, the PUFA value increased from 4.1% for CO to 36.5% for S-100%. PUFAs and MUFAs are known to lower plasma cholesterol levels and thus prevent cardiovascular disease [42] Reducing SFAs is extremely important from a consumer perspective, as SFAs are considered to be fatty acids that are harmful to human health. International organizations such as the USDA (United States Department of Agriculture), FAO (Food and Agriculture Organization of the United Nations), and EFSA (European Food Safety Agency) recommend the lowest possible intake of SFA [43,44,45]. The use of a fat substitute reduced SFA by 22.6% for the grilled burgers (50.3 and 27.7% for CO and S-100% respectively, *p* < 0.05). The differences between the fatty acid profile of raw and grilled burgers are due to the mechanisms occurring during the thermal processing of the burger, during which water loss, fat loss, oxidation, hydrolysis, or polymerization of triacylglycerol molecules occur [42].

In addition to the assessment of the fatty acid profile, the nutritional quality of the acids can be evaluated using indices such as PUFA/SFA, Al, Tl, and h/H. These indices take into account the different (both suppressing and promoting) effects of the fatty acids on these processes. SFAs, which are thrombogenic, are not the same as atherogenic SFAs, as well as MUFAs, PUFAs show different degrees of protection against cardiovascular disease (thrombosis and atherosclerosis) [46]. It is recommended to consume products with a low atherogenicity and thrombogenicity index and the highest possible h/H ratio [46]. However, a PUFA:SFA ratio >0.45 is recommended, as a lower value may contribute to cardiovascular disease [42,47]. The effect of using fat substitutes on nutritional indices is shown in Figure 7. In the experiment conducted, the ratio was significantly increased (*p* < 0.05); for CO it was 0.06 while for the reformulations used it ranged from 0.38 (S-25%) to 1.3 (S-100%). The ratio increased with increasing fat substitution. Moreover, on day 0, the control variant in raw burgers had Tl, Al, h/H amounting to 2.03, 0.9, and 1.29, respectively. The use of the fat substitute with encapsulated açai oil improved the health value of the beef burgers with Tl, Al, h/H amounting to 0.54, 0.25, and 4.65, respectively. The values of the indices increased significantly (*p* < 0.05) with increasing animal fat replacement. A similar trend was also observed for the grilled burgers, where the variant S-100% had the best nutritional parameters. Similar Al values (0.24) were found in the study by Rodríguez-Carpena et al. [48], who used avocado oil, sunflower oil, and olive oil as fat substitutes. Depending on the fat substitute and the type of vegetable fat source used, they affect the health index scores to different degrees [32,42,49].

### 2.4. Sensory Evaluation of Burger Beef

As fat is an essential component of processed meat products, it plays an important role in the design of many parameters. Therefore, an assessment of consumer acceptance should be considered for products in which fat substitutes are used [30]. The results of consumer evaluation of grilled burgers with fat substitutes on day 0 and day 7 are presented in Table 3. The consumer ratings of burgers on day 0 and day 7 ranged from 4.0 to 8.6 points on an unstructured (linear) scale, with 0 meaning unacceptable and 10 points meaning a very acceptable product. On day 0, the results showed that the reformulated burgers did not differ (*p* < 0.05) in parameters such as color, flavor and juiciness. On the other hand, the consumer ratings showed significant differences (*p* < 0.05), with appearance, taste, texture, and overall acceptability, decreasing with increasing animal fat substitution on day 0. However, consumers rated both the control variant and the burgers with fat replacement worse after 7 days of storage than on day 0. This effect indicates that with increasing storage time, changes occur that lead to a loss of product quality [33]. On day 0, consumers rated the burgers CO, S-25%, and S-50% with >6.6 points for each parameter analyzed. Nevertheless, no reformulation was unacceptable to consumers during the analysis. The high scores of the control variant (>8.0) in the analysis of appearance, color, flavor, taste, juiciness, texture, and overall acceptability confirm the importance of the type of fat in a burger and how challenging it is to substitute animal fat with a fat replacement to fully meet consumer needs [29].

### 2.5. Correlation between the Chemical and Physical Parameters Analyzed in Grilled Burgers

Correlation analysis (Table 4) showed that the evaluated parameters of the consumer analysis (appearance, color, flavor, taste, juiciness, texture, and overall acceptability) are highly correlated (*p* < 0.05), with values ranging from 0.80 to 0.97. Consumer rating is influenced by many coupled attributes of the product being evaluated. Any change in one sensory attribute can lead to a different perception of other attributes and the overall quality [50]. Therefore, high significance was found in the correlation analysis not only for the consumer evaluation parameters but also for the physicochemical analyses (SFA, MUFA, PUFA, hardness, springiness, L* a* b*). The analysis also revealed a highly significant negative correlation between SFA, MUFA and the content of volatile compounds belonging to the ketone group (−0.78, −0.75, *p* < 0.05, respectively). In contrast, a positive correlation was found for PUFA and ketone compounds (0.93, *p* < 0.05). Moreover, a significant negative correlation was found for SFA and PUFA content. This is probably related to the oxidation of fatty acids, under the influence of which the double bonds are reduced until saturation. The correlation analysis also showed a positive correlation between hardness (TPA analysis) and elasticity (TPA analysis). Correlation analysis of the color L* showed it had a high positive correlation with the color parameters a* and b* (0.86, 0.73, *p* < 0.05, respectively). Similar relationships were found in a previous study by [33]. Moreover, the color parameter pH is negatively correlated with volatile compounds from the alcohol group (−0.8, *p* < 0.05). 

## 3. Material and Methods

### 3.1. Preparation of the Fat Substitute

A freeze-dried hydrogel emulsion based on konjac flour and sodium alginate with encapsulated açai oil (PURO PURA, Brazil’s Finest). and the addition of flaxseed meal flour was used as a fat substitute. The hydrogel emulsion was prepared according to a method previously developed by the authors [33]. Briefly, gelation involved dissolving 2 g of sodium alginate (agnex 1999) and 0.86 g of konjac flour (green essence) in 100 mL of H_2_O at 60 °C with constant stirring until a clear solution was obtained. The prepared emulsion (2:1 oil: H_2_O) was homogenized in the biocomposite obtained. Then, defatted flaxseed flour (Oleofarm, LenVitol) was added to crisp up the freeze-dried product (5 g flaxseed flour/95 g hydrogel emulsion). The resulting hydrogel with encapsulated oil (31%) was freeze-dried.

### 3.2. The Preparation and Packaging of Beef Burgers

The characteristics of the raw materials used in the experiment (base composition of the meat, fatty acid profile of beef tallow and açai oil, and color of fat substitute) are given in Table 5. The burgers were prepared according to the reformulation shown in Table 6. The beef (neck muscles, from Zakłady Mięsne Wierzejki Sp. z o.o., Płudy, Poland) without excess connective tissue and fat was minced. The beef and beef tallow were then ground separately using a meat grinder with an Ø 8 mm plate (PI-22-TU-T, Edesa, Mondragón, Spain). The control variant was a burger with beef tallow added. All the indicated ingredients of the reformulation were mixed in the specified proportions to form burgers with the following dimensions: 120 ± 1 g, Ø 10 cm, 25 mm thick. The formed burgers were then placed on trays (polyethylene terephthalate-PET) measuring 137 × 187 × 50 mm. The burgers were packed in MAP (80 O_2_, 20% CO_2_), using a 35-µm-thick PSF film (PSF35ZAC, PolTechPack Sp. z o.o. The packaged burgers were stored for 7 days at 4 °C ± 1 °C. Analyses such as color, pH, fatty acid profile, volatile compound profile, and TBARS were carried out on raw burgers on days 0 and 7. On a specified day of analysis, the burgers were grilled (Grill Machines, model S-165 K, Silex, Hamburg, Germany), and heated to 210 °C (top plate) and 190 °C on the bottom plate until 75 °C was reached at the geometric center of each burger (measured using a thermocouple). The burgers were then cooled to 25 °C. The grilled burgers were used for tests, including color, cooking loss, texture profile analysis (TPA), volatile compound profile analysis, fatty acid profile analysis, and consumer evaluation on days 0 and 7 of storage. The experimental setup involved two biological replicates.

### 3.3. Chemical Analyses

#### 3.3.1. Thiobarbituric Acid Reactive Substances Analysis (TBARS)

The extraction method and method for the determination of lipid oxidation by Brodowska et al. [51] was applied with a minor modification. The absorbance of the resulting reaction color complex was measured using a UV-VIS spectrophotometer (Shimadzu UV-1800; Shimadzu, Kyoto, Japan). The resulting lipid oxidation is presented as milligrams of malondialdehyde (MDA)/100 g sample. 

#### 3.3.2. Analysis of the Fatty Acid Profile and Health Indicators

Fatty acid profile analysis was performed on beef tallow, açai oil, raw burger, and grilled burgers after 0 and 7 days of storage. Lipids were methylated using the method described by Wojtasik-Kalinowska et al. [52] and Heck et al. [53] that was slightly modified. The resulting mixture of fatty acid methyl esters was analyzed using a gas chromatograph (Shimadzu GC-2010, Shimadzu, Kyoto, Japan) with a flame ionization detector (FID). The chromatograph was equipped with a Zebron ZB-FAME column (GC Cap. Column, 60 m × 0.25 mm × 0.2 um, Ea). The initial column temperature was 100 °C, held for 3 min, then was increased gradually to 240 °C at a rate of 2.5 °C/min. The final temperature was held for 10 min. For the identification of FAME, the FAME Mix-37 pattern was used (Supelco, TraceCERT^®^, EC:200-838-9, SKU: CRM47885). The results obtained were presented as a fatty acid profile. In addition, thrombogenicity (TI) and atherogenicity (Ai) were determined according to the method described by Ulbricht and Southgate [46] and the hypocholesterolemic/hypercholesterolemic (h/H) ratio was calculated according to Fernández et al. [54] according to the following formula.
$$\begin{array}{c} \mathrm{TI}=\frac{C14:0+C16:0+18:0}{\left(0.5\times \Sigma \mathrm{MUFA}\right)+\left(0.5\times \;\Sigma n-6\right)+\left(3\times \Sigma n-3\right)+\left(\frac{\Sigma n-3}{\Sigma n-6}\right)}\\ \mathrm{AI}=\frac{C12:0+\left(4*\;C14:0\right)+16:0}{\left(\Sigma \mathrm{PUFA}\;n-3\right)+\left(\Sigma \mathrm{PUFA}\;n-6\right)+\left(\Sigma \mathrm{MUFA}\right)}\\ \frac{h}{H}=\frac{C18:1n9+\Sigma \;\mathrm{PUFA}\;}{C14:0+C16:0} \end{array}$$

#### 3.3.3. Analysis of the Volatile Compounds

Analysis of volatile compounds was performed using a Heracles II e-nose instrument (Alpha MOS Co., Toulouse, France), which employed gas chromatography with a flame ionization detector and a retention index counting application with AroChemBase library (AlphaSoft software v12.42, Alpha MOS Co., Toulouse, France). The gas chromatograph was equipped with two capillary columns of different polarity, DB-1701, and DB-5, with 10 m × 0.18 mm ID × 0.4 μm film thickness. Analysis was conducted according to the methodology reported by Górska-Horczyczak et al. [55]. Calibration was performed on a standard mixture of C6-C16 alkanes (Restek, ANCHEM Plus, Warsaw, Poland).

### 3.4. Physical Analyses

#### 3.4.1. Scanning Electron Microscope (SEM) Analysis

The morphology of the freeze-dried hydrogel emulsion with açai oil capsules was analyzed using a high-resolution, scanning electron microscope JEOL JSM–7500 F, Field Emission Scanning Electron (Akishima, Tokyo, Japan) using the methods described by Nowak et al. [56].

#### 3.4.2. Color and pH Analysis

The color was evaluated on grilled and raw burgers after 0 and 7 days of storage. The color measurement was performed on the entire surface, at 5 different locations for one burger. The analysis measurement was performed in two batches. Analysis was performed using a Minolta CR-400 chromameter with a glass cell (10 mm optical path) with the CA-A98 attachment (Konica Minolta Inc., Tokyo, Japan) and D65 illuminant (color temperature: 6500 K), as well as a standard observer (2°) in the CIE L* a* b* system. The diameter of the measuring head was 8 mm. For calibration, the white standard plate was used (L* = 98.45, a* = −0.10, b* = −0.13). In addition, the total color difference value (ΔE) was calculated according to the formula by Altmann et al. [23]. The pH was measured using a digital pH meter (FiveEasy F20, Mettler Toledo, Greifensee, Switzerland).

#### 3.4.3. Texture Profile Analysis (TPA) and Cooking Loss

The texture profile analysis (TPA) of grilled beef burgers after 0 and 7 days of storage at 4 ± 1 °C was performed according to the method described by Hanula et al. [33] in a double compression cycle to the point of 50% reduction of their initial height (relaxation time 3 s). The analysis was performed using an Instron 5965 (Instron, Norwood, MA, USA) testing machine with a 500 (N) load cell connected to Bluehill^®^ 2 software (Norwood, MA, USA). For TPA, we used an Instron S5402A probe (dimensions: 57.32 mm (2.26 in) diameter × 86 mm (3.39 in) long) with a surface area of 2580.49 mm^2^. The samples had an area of 506.70 mm^2^ (25.4 mm diameter (1.00 in)). The parameters such as hardness (N), springiness (-), and cohesiveness (-) were calculated using the method described by Półtorak et al. [57]. 

Cooking loss analysis was performed after 7 days of burger storage at 4 °C. The cooking loss value was expressed as a percentage and calculated using the following formula:cooking loss % = (BR − BG)/BR × 100

BR—raw burger weight after 7 days of storageBG—grilled burger weight after 7 day of storage

### 3.5. Sensory Evaluation of Grilled Beef Burger

The sensory evaluation of grilled beef burgers was performed using an unstructured linear scale of a 10 cm in length, on a scale from ‘‘no intensity’’ to ‘’very high intensity’’ [58]. The sensory panel consisted of 11 experienced panelists (5 women and 6 men, aged 31–45 years). Panelists passed the validation process and met the criteria of ISO 11132 [59]. Each trait analyzed was scored three times, and grilled burgers were provided in coded cardboard trays in random order. Each panelist received water to cleanse the palate between tastings. The analog scale scores obtained were converted to numerical values from 0 to 10 units. Grilled burgers were evaluated on day 0 and after 7 days of storage.

### 3.6. Statistical Analyses

The statistical analyses of reformulated beef burger results were performed using the Statistical Analysis System software version 13.3 (StatSoft, Tulsa, OK, USA). The normality of the data distribution and the homogeneity of variance was previously checked using the Shapiro–Wilk test. A one-way ANOVA followed by the Tukey’s test was applied for the analysis of pH, color, TBARS, fatty acid profile (SFA, MUFA, PUFA), health indicators (h/H, TI, AI), TPA, cooking loss, and consumer evaluation. The level of significance was set at α ≤ 0.05. Moreover, the strengths of the relationships between fatty acid profile (SFA, MUFA, and PUFA), texture parameters (springiness, hardness), cooking loss, pH, color parameters (L*, a*, b*), and consumer parameter (appearance, color, flavor, taste, juiciness, texture, overall acceptability) methods were determined using Pearson correlation coefficients. For all analyses, 95% confidence intervals were established. The AlfaSoft package with statistical quality control was used for the evaluation of the chromatographic fingerprint and the comparative analysis of volatile compounds

## 4. Conclusions

The aim of this study was to analyze the effects of using a freeze-dried hydrogel emulsion with encapsulated açai oil as a fat substitute. SEM analysis showed that the freeze-drying process had no destructive effect on the encapsulated oil. Moreover, the fat replacement had a positive effect on improving the nutritional value of the burger as it reduced SFA and enriched PUFA. Furthermore, the nutritional indices of both the grilled and raw burger were significantly improved. For example, the atherogenicity and thrombogenicity indices were significantly reduced by 0.5 and 1.2, respectively (*p* < 0.05). In contrast, the h/H ratio improved by 2.6. Furthermore, the fat substitute significantly reduced all the texture parameters analyzed (hardness, springiness, cohesiveness). In addition, the freeze-dried hydrogel reduced cooking loss for all variants used. Analysis of the volatile compound profile showed that, after 7 days of storage, the beef fat produced the most ketones, while the change in distances (organoleptic unit) in the raw burger was lowest in S-100% and increased with an increasing proportion of animal fat in the burger. The correlation analysis (*p* < 0.05) showed a high significance between the analyzed parameters and consumer evaluation. Despite the influence of many correlating features on the consumer rating, burgers with fat substitutes S-25%, and S-50% were rated above six points, so the consumer accepted the fat substitute. The study showed that the fat substitute freeze-dried hydrogel with encapsulated açai oil can successfully replace animal fat by up to 50%, improving the health benefits of the beef burger.

## Figures and Tables

**Figure 1 molecules-27-03700-f001:**
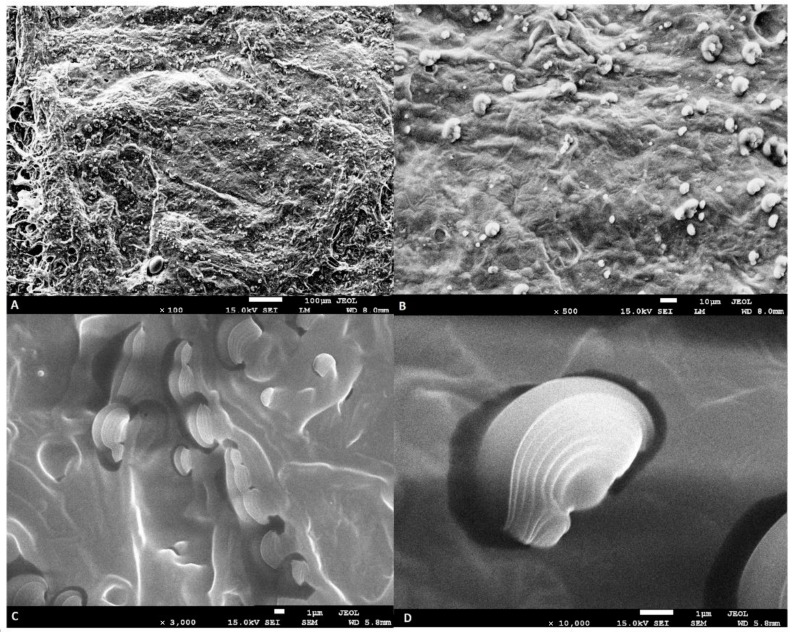
SEM analysis of freeze-dried emulsion hydrogels with encapsulated açai oil (31%), (**A**,**B**)—distribution of nanocapsules; (**C**,**D**)—multilayer nanocapsules.

**Figure 2 molecules-27-03700-f002:**
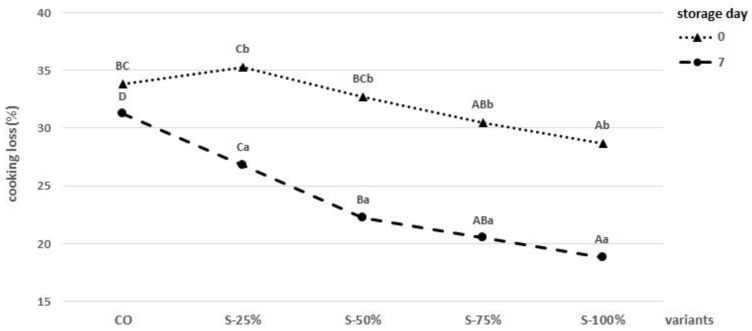
Analysis of cooking losses in grilled burgers with fat replacement after 0 and 7 days of storage. ^A–D^—means with different letters showing a significant effect of the treatment group in the same day of storage, *p* < 0.05. ^a,b^—means with different letters showing a significant effect of storage time in each treatment group, *p* > 0.05.

**Figure 3 molecules-27-03700-f003:**
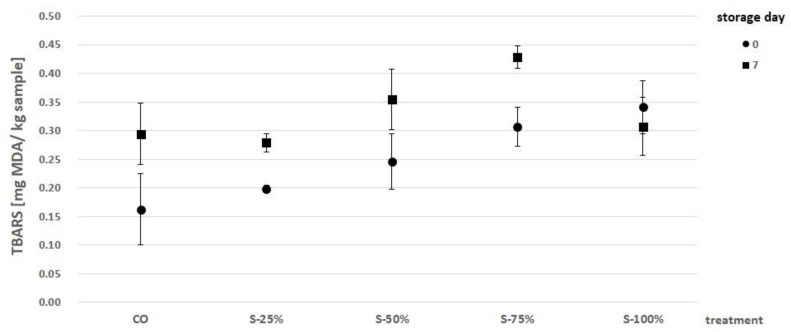
Analysis of lipid oxidation (TBARS) in burgers with substituted fat after 0 and 7 days of storage. CO-control with tallow (0% fat substitute), S-25%, S-50%, S-75%, S-100% means, respectively, replacing beef fat with 25, 50, 75, and 100% substitute.

**Figure 4 molecules-27-03700-f004:**
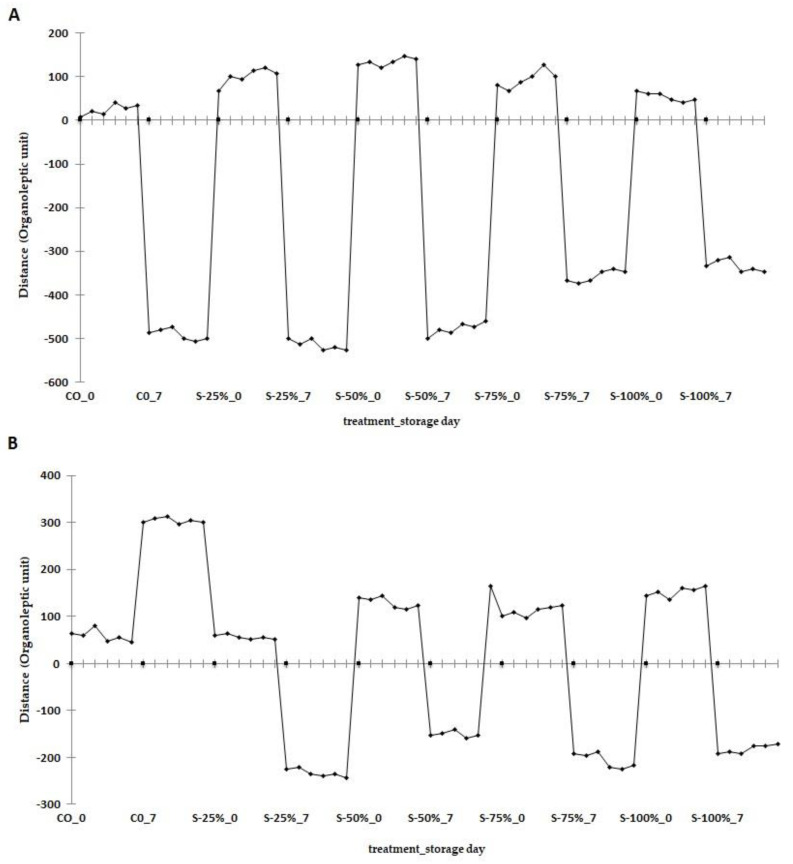
Changes in the volatile compounds profile of raw burgers (**A**); grilled burgers (**B**); after 0, 7 days of storage (4 ± 1 °C); CO-control with tallow (0% fat substitute), S-25%, S-50%, S-75%, S-100% means respectively replacing beef fat with 25, 50, 75, and 100% substitute.

**Figure 5 molecules-27-03700-f005:**
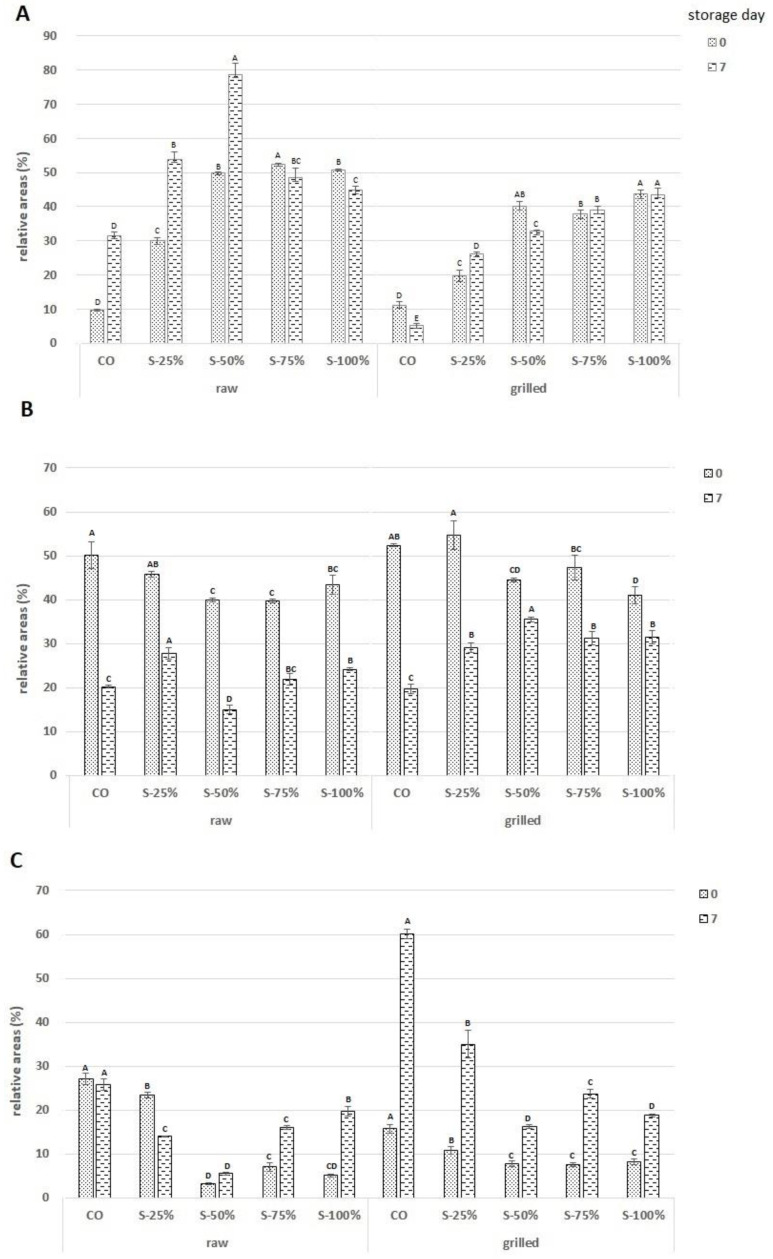
The relative peak areas (%) of the three main groups of volatile compounds: ketones (**A**), alcohols (**B**), and aldehydes (**C**) in beef burgers with substituted fat after 0 and 7 days of storage; CO-control with tallow (0% fat substitute), S-25%, S-50%, S-75%, S-100% means respectively replacing beef fat with 25, 50, 75, and 100% substitute. ^A–E^—means with different letters showing a significant effect of the treatment group in the same day of storage, *p* < 0.05.

**Figure 6 molecules-27-03700-f006:**
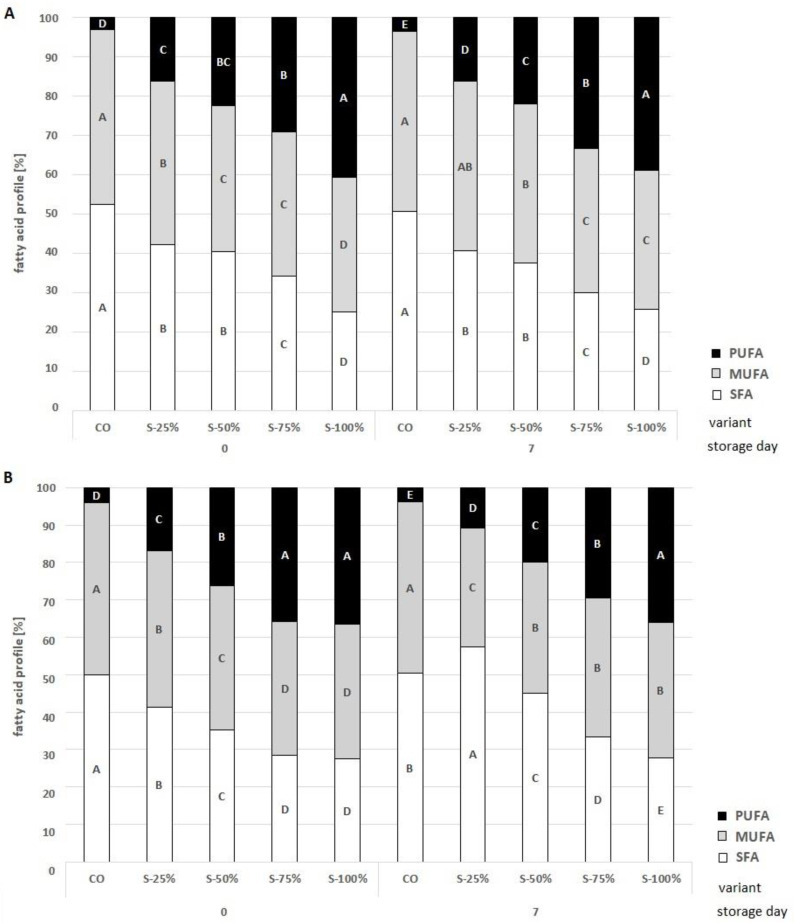
Effects of a fat substitute (freeze-dried hydrogel emulsion with encapsulated açai oil) on the fatty acid profile of raw (**A**) and grilled (**B**) burgers at 0 and 7 days of storage in cold conditions; CO-control with tallow (0% fat substitute), S-25%, S-50%, S-75%, S-100% means respectively replacing beef fat with 25, 50, 75, and 100% substitute. ^A–E^—means with different letters showing a significant effect of the treatment group in the same day of storage, *p* < 0.05.

**Figure 7 molecules-27-03700-f007:**
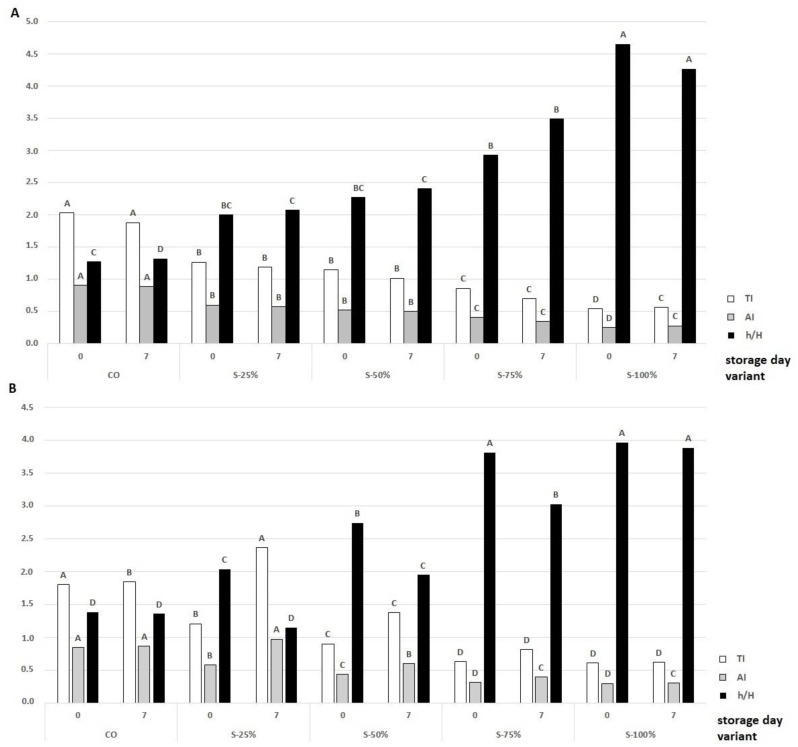
Effects of a fat substitute (freeze-dried hydrogel emulsion with encapsulated açai oil) on the nutritional indexes AI, TI, and h/H of raw (**A**) and grilled (**B**) burgers at 0 and 7 days of storage in cold conditions; CO-control with tallow (0% fat substitute), S-25%, S-50%, S-75%, S-100% means respectively replacing beef fat with 25, 50, 75, and 100% substitute. ^A–D^—means with different letters showing a significant effect of the treatment group in the same day of storage, *p* < 0.05.

**Table 1 molecules-27-03700-t001:** Effect of the fat substitute used (freeze-dried hydrogel emulsion with encapsulated açai oil) on the color parameter (L*, a*, b*) and pH at 0 and 7 days of storage for both raw and grilled burgers.

	Day 0				Day 7			
	L*	a*	b*	pH	L*	a*	b*	pH
raw								
CO	42.15 (3.56)	21.26 (3.10) ^b^	10.96 (1.3) ^Ab^	5.38 (0.02) ^Aa^	47.19 (3.27) ^B^	14.01 (1.66) ^Aa^	7.83 (0.78) ^Aa^	5.67 (0.05) ^Ab^
S-25%	42.26 (2.91)	22.17 (1.69) ^b^	12.03 (1.31) ^ABb^	5.50 (0.02) ^Ba^	43.89 (4.18) ^AB^	17.24 (1.62) ^Ba^	9.87 (0.85) ^Ba^	5.81 (0.13) ^Bb^
S-50%	42.75 (2.46)	22.43 (0.96) ^b^	11.75 (1.16) ^ABb^	5.51 (0.04) ^Ba^	41.91 (2.23) ^A^	19.05 (2.60) ^BCa^	9.92 (1.51) ^Ba^	5.68 (0.03) ^Ab^
S-75%	42.71 (2.17)	22.11 (2.13) ^b^	11.67 (1.32) ^ABb^	5.56 (0.02) ^Ca^	42.73 (4.15) ^A^	19.06 (1.45) ^BCa^	10.30 (0.99) ^BCa^	5.69 (0.05) ^ABb^
S-100%	43.33 (2.59)	23.16 (1.36) ^b^	12.99 (0.89) ^Bb^	5.56 (0.02) ^Ca^	41.77 (2.30) ^A^	20.05 (1.18) ^Ca^	11.46 (1.28) ^Ca^	5.67 (0.03) ^Ab^
SEM	0.381	0.283	0.189	0.016	0.534	0.388	0.226	0.016
grilled								
CO	30.75 (2.37) ^Ba^	6.76 (0.40) ^A^	6.8 (1.01) ^a^		37.76 (2.47) ^b^	6.38 (0.55) ^A^	8.82 (0.90) ^Ab^	
S-25%	26.27 (2.19) ^Aa^	6.78 (1.40) ^A^	6.80 (1.50) ^a^		34.96 (2.73) ^b^	6.72 (0.51) ^A^	9.07 (0.97) ^Ab^	
S-50%	28.36 (3.03) ^ABa^	7.75 (1.16) ^AB^	8.24 (1.53) ^a^		37.39 (2.65) ^b^	7.86 (0.77) ^B^	10.09 (0.96) ^ABb^	
S-75%	29.14 (2.3) ^ABa^	8.05 (1.18) ^AB^	7.89 (1.58) ^a^		37.76 (2.90) ^b^	8.04 (0.84) ^BC^	10.31 (1.79) ^ABb^	
S-100%	29.74 (1.5) ^Ba^	8.47 (1.17) ^B^	8.33 (1.34) ^a^		38.32 (2.74) ^b^	8.89 (1.15) ^C^	10.86 (1.23) ^Bb^	
SEM	0.382	0.180	0.214		0.403	0.170	0.198	

The numbers in brackets are standard erros.^A–C^—means with different letters showing a significant effect of the treatment group in the same day of storage, *p* < 0.05. ^a,b^—means with different letters showing a significant effect of storage time in each treatment group, *p* > 0.05.

**Table 2 molecules-27-03700-t002:** Influence of freeze-dried hydrogel emulsions with the encapsulation açai oil as a fat substitute on the texture parameters (TPA) of grilled burgers at 0 and 7 storage day.

Storage Day	Variants	Hardness (N)	Springiness (-)	Cohesiveness (-)
0	CO	75.82 (5.07) ^Ca^	0.53(0.06) ^C^	0.39 (0.03) ^B^
	S-25%	31.44 (2.31) ^Ba^	0.48 (0.03) ^BC^	0.36 (0.02) ^Bb^
	S-50%	20.64 (7.07) ^BA^	0.39 (0.07) ^B^	0.33 (0.02) ^Bb^
	S-75%	10.29 (0.47) ^Aa^	0.2 (0.01) ^A^	0.23 (0.01) ^Aa^
	S-100%	10.21 (0.68) ^Aa^	0.26 (0.04) ^A^	0.25 (0.03) ^A^
7	CO	102.36 (1.94) ^Db^	0.52 (0.01) ^B^	0.38 (0) ^C^
	S-25%	44.83 (3.25) ^Cb^	0.4 (0.04) ^AB^	0.3 (0.02) ^Ba^
	S-50%	27.36 (4.57) ^B^	0.38 (0.05) ^AB^	0.28 (0.03) ^ABa^
	S-75%	20.57 (3.13) ^BAb^	0.34 (0.09) ^AB^	0.29 (0.03) ^ABb^
	S-100%	18.63 (1.07) ^Ab^	0.27 (0.06) ^A^	0.24 (0.02) ^A^
SEM		5.366	0.021	0.011

The numbers in brackets are standard error. ^A–D^ mean values between variants on the same storage day with different letters indicate significant difference. ^a,b^ mean values on the same variants. between storage day with different letters indicate significant difference.

**Table 3 molecules-27-03700-t003:** Analysis of the effect of substitute fat on the sensory attributes measured on the 0 and 7 storage day.

Storage Day	Treatment				
	CO	S-25%	S-50%	S-75%	S-100%
	Appearance				
0	8.6 (1.26) ^A^	7.6 (1.47) ^AB^	7.0 (0.92) ^AB^	6.4 (2.47) ^AB^	5.5 (2.30) ^B^
7	8.2 (0.96) ^A^	7.9 (0.98) ^A^	6.0 (1.34) ^B^	5.7 (1.34) ^B^	5.2 (2.28) ^B^
	Color				
0	8.1 (1.40)	7.6 (1.19)	6.7 (1.33)	6.6 (1.73)	6.5 (1.82)
7	8.0 (0.76) ^A^	7.4 (1.64) ^AB^	5.9 (2.21) ^B^	5.9 (1.36) ^B^	5.5 (1.29) ^B^
	Flavor				
0	8.4 (1.46)	8.0 (1.34)	7.0 (1.71)	6.7 (1.94)	6.5 (2.62)
7	7.9 (1.20) ^A^	7.4 (1.59) ^AB^	6.0 (1.50) ^AB^	5.7 (1.70) ^B^	5.4 (1.78) ^B^
	Taste				
0	8.5 (1.30) ^A^	7.9 (1.24) ^AB^	6.6 (1.57) ^ABC^	5.9 (2.23) ^BC^	5.4 (1.59) ^C^
7	7.3 (1.41) ^A^	6.8 (1.30) ^A^	5.0 (1.84) ^B^	4.8 (1.94) ^B^	4.7 (1.79) ^B^
	Juiciness				
0	8.0 (1.80)	7.5 (1.87)	7.3 (1.47)	6.2 (1.77)	6.3 (1.63)
7	6.9 (2.16)	6.6 (1.41)	5.7 (2.01)	5.1 (1.30)	5.1 (1.64)
	Texture				
0	8.2 (1.80) ^A^	7.4 (2.26) ^A^	6.6 (1.58) ^AB^	4.7 (1.94 ^BC^	4.0 (2.57) ^C^
7	6.7 (2.13)	6.5 (2.06)	5.4 (2.02)	4.9 (1.94	4.3 (1.66)
	Overall acceptability				
0	8.3 (1.55) ^A^	7.6 (1.69) ^AB^	6.8 (1.76) ^AB^	5.6 (2.21) ^BC^	4.5 (2.26) ^C^
7	7.4 (1.40) ^A^	6.7 (1.61) ^AB^	5.1 (1.42) ^B^	5.0 (1.45) ^B^	4.9 (1.72) ^B^

The numbers in brackets are standard error ^A–C^ mean values between variants on the same storage day with different letters indicate significant difference. CO-control with tallow (0% fat substitute), S-25%, S-50%, S-75%, S-100% means respectively replacing beef fat with 25, 50, 75, and 100% substitute.

**Table 4 molecules-27-03700-t004:** Analysis of correlations between the fatty acid profile (SFA, MUFA, and PUFA), color parameters (L*, a*, b* and BI) and WHC, pH, weight and cooking loss, TBARS, and TPA parameters (springiness, hardness) in grilled burgers.

	SA	SC	SF	STa	SJ	STe	SOa	FS	FM	FP	TT	TS	L*	a*	b*	pH	EAd	EAk	EK
SA	1																		
SC	0.92 *	1																	
SF	0.95 *	0.96 *	1																
STa	0.94 *	0.94 *	0.98 *	1															
SJ	0.83 *	0.87 *	0.92 *	0.92 *	1														
STe	0.91 *	0.80 *	0.86 *	0.90 *	0.84 *	1													
SOa	0.96 *	0.88 *	0.93 *	0.96 *	0.87 *	0.97 *	1												
FS	0.81 *	0.6	0.64 *	0.62	0.41	0.71 *	0.69 *	1											
FM	0.6	0.67 *	0.65 *	0.69 *	0.65 *	0.66 *	0.70 *	0.26	1										
FP	−0.90 *	−0.75 *	−0.78 *	−0.77 *	−0.58	−0.84 *	−0.83 *	−0.94 *	−0.58	1									
TT	0.79 *	0.75 *	0.69 *	0.67 *	0.49	0.68 *	0.72 *	0.75 *	0.73 *	−0.90 *	1								
TS	0.82 *	0.7 *	0.75 *	0.78 *	0.66 *	0.89 *	0.85 *	0.79 *	0.72 *	−0.93 *	0.82 *	1							
L*	−0.27	−0.37	−0.5	−0.50	−0.66 *	−0.29	−0.34	0.19	−0.15	−0.10	0.26	−0.01	1						
a*	−0.95 *	−0.85 *	−0.88 *	−0.85 *	−0.71 *	−0.86 *	−0.88 *	−0.87 *	−0.55	0.93 *	−0.80 *	−0.86 *	0.13	1					
b*	−0.66 *	−0.75 *	−0.84 *	−0.83 *	−0.88 *	−0.63	−0.69 *	−0.21	−0.51	0.36	−0.24	−0.4	0.86 *	0.54	1				
pH	−0.28	−0.38	−0.47	−0.53	−0.65 *	−0.41	−0.44	0.20	−0.61	0.05	−0.05	−0.23	0.73 *	0.09	0.76 *	1			
EAd	0.42	0.39	0.25	0.19	0.02	0.25	0.28	0.60	0.30	−0.61	0.78 *	0.48	0.63	−0.55	0.26	0.54	1		
EAk	0.15	0.2	0.34	0.40	0.50	0.29	0.29	−0.22	0.18	0.12	−0.32	0.01	−0.86 *	0.00	−0.75 *	−0.83 *	−0.8 *	1	
EK	−0.88 *	−0.84 *	−0.83 *	−0.82 *	−0.62	−0.80 *	−0.84 *	−0.78 *	−0.75 *	0.93 *	−0.93 *	−0.88 *	−0.02	0.92 *	0.48	0.18	−0.62	0.05	1

^*^*p* ≤ 0.05, SA—appearance, SC—color, SF-flavor, STa—taste, SJ—juiciness, STe—texture, SOa—overall acceptability, FS—SFA, FM—MUFA, FP—PUFA, TT—hardness, TS—springiness, EAd—aldehyde, EAk—alcohol, EK—ketone.

**Table 5 molecules-27-03700-t005:** Basic composition of the substrates used in the experiment.

Fatty Acid Profile (%)	Açai Oil	Tallow
C14:0	0.08	3.57
C14:1 n5 c-9	0.00	1.19
C15:0	0.01	0.36
C16:0	8.27	26.91
C16:1 n7 c-9	0.08	3.79
C17:0	0.06	0.85
C17:1 n7 c-10	0.03	0.63
C18:0	3.38	15.36
C18:1 n9 c-9	28.73	44.13
C18:2 n6 c-9,12	54.51	1.71
C18:3 n3 c-9,12,15	2.95	0.24
C20:0	0.32	0.11
C18:2 c-9, t-11	0.00	0.43
C22:0	0.53	0.02
C24:0	0.18	0.00
SFA	13.35	47.38
MUFA	28.84	49.76
PUFA	57.81	2.86
Primary composition	meat	
Protein	19.6%	
Moisture	68.54%	
fat	8.21%	
color	Substitute fat	
L*	54.87	
a*	25.11	
b*	16.34	

**Table 6 molecules-27-03700-t006:** Formulations of beef burger with substitute fat (konjac flour, sodium alginate, linseed flour, encapsulated açai oil), CO—control (0% substitute fat), S-25%—25% substitute fat, S-50%—50% substitute fat, S-75%—75% substitute fat, S-100%—100% substitute fat. These proportions represent the ingredients of a 120 g burger.

Treatment Identification	Substitute Fat (%)	Beef Meat (g)	Tallow (g)	Freeze-Dried Emulsion Hydrogel (g)
CO	0	96	24	0
S-25%	25	96	18	6
S-50%	50	96	12	12
S-75%	75	96	6	18
S-100%	100	96	0	24

All are formulated with 1.4% salt and 0.35% pepper.

## Data Availability

Available from the authors.

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
