# Peer review of "Quality of Beef Burgers Formulated with Fat Substitute in a Form of Freeze-Dried Hydrogel Enriched with Açai Oil"

_molecules, 2022, doi:10.3390/molecules27123700_

Round 1
Reviewer 1 Report
The research was well planned and studied. It has important results both in terms of technology and its effects on human health. The authors should clearly reveal the differences between this research and their previous research, especially in the beef burger section.
There is a mismatch as a unit in the values given between lines 204-211 regarding TBARS values. In case of burgers, if there is 2.92 mg of MDA in 100 g of burger, it is 29.9 MDA in 1 kg. That is, it is higher than the threshold value. I suggest that this issue be elaborated and clarified and rechecked.
The meaning of the +/- values given in the tables should be given as footnote under tables. Likewise, standard errors should be shown on the shape bars. Further discussion is required.
Author Response
Dear Reviewer,
Thank you very much for your time and valuable comments. I have addressed any comments that were made in the form of tracking changes and the responses to the comment are provided in the attached file.
Kind Regards,

Reviewer 2 Report
Over the past few years Texture Profile Analysis testing has been the cause of much concern. In general, TPA is a very popular method of testing, as it provides very quick calculation of parameters which are 'believed to correlate with sensory analysis'.
The following is a set of points to consider when choosing TPA as your test procedure:
#1 Size of Compression Probe versus Sample
When the probe is larger than the sample, the forces registered are largely due to uniaxial compression. However, when the opposite is true, the forces derive largely from puncture, a combination of compression and shear. Various papers throughout the decades of using TPA have reported the use of probes both larger and smaller than the test samples. Early papers on TPA report the use of puncture probes, but in 1968 Prof. Malcolm Bourne was the first to adopt true uniaxial compression to perform TPA tests. Generally speaking, most recent work done on TPA uses compression probes of the same size as or larger than the sample size, so that the forces registered in such TPA tests are largely due to uniaxial compression forces and the whole of the sample piece is tested. Unfortunately, the manuscript lacks this important information on the sample size (surface area) and the size of the probe.
#2 Extent of Deformation
Another area of abuse is the degree of compression. Often when presumably limited by force capacity we find that results are shown for compression to, for example, 30%. If the purpose of testing is to imitate the highly destructive process of mastication in the mouth, as in the TPA's origins, deformation values to break the sample must be reached. We need info on this as well.
#3 When setting a sensory panel, first step is its validation to confirm that the panel can work and be used in sensory studies. Also, validation of training (prior to sensory analysis), may result in exclusion of panelists due to discriminating problems. No information is provided regarding validation methods used and resulting activities applying to this sensory panel (https://doi.org/10.1111/jtxs.12616 ).
#4 Besides validating the panel (prior to sensory analysis), it is also important to assess panelists performance. Criteria for evaluating the attributes of a trained sensory panel and evaluation of the panel performance cover the following aspects: (i) is the panel capable of showing products differences / discriminate in-between samples; (ii) are the scores of panelists reliable (in-between replicates and over time in case of evaluating products over time); (iii) are results valid in terms of visible consensus between panelists and scoring in a similar way; and (iv) are they able to specify specific sensory attributes and sensations. Standard ISO 11132 outlines all four criteria as of equal importance: discriminability (linked with differences between products), homogeneity (consensus of the panel), repeatability (within sessions), and reproducibility (between session) as well as two-way ANOVA for panel performance (discrimination, homogeneity and repeatability) and one-way ANOVA for assessor performance (discrimination and repeatability). No information is provided regarding assessment of the panel performance used in this study.
Without the information on sensory panel validation and assessment of the panel performance we can not trust the results obtained by the panel.
#5 The need for standardized set of minimum reportable parameters for instrumental meat color evaluation still remains to be identified and incorporated in peer-reviewed journals guidelines for authors, as it was the case a decade ago. In the most recent review regarding meat color https://doi.org/10.1016/j.cofs.2021.02.012 the authors are proposing that all manuscripts containing instrumental color data must report of instrumental details that include (at least) the information on: instrument and its calibration, illuminant, aperture size, degree of observer and number of readings per sample. Unfortunately, this manuscript missed to report some of the parameters as well.
#6 Finally, the magnitude of color difference between the two methods used is best represented by the total color difference value (ΔE). The info about how it is calculated and about its threshold for human meat-color difference detection is described in https://doi.org/10.1016/j.meatsci.2022.108766
Author Response

(The authors gave the same response as above.)

Round 2
Reviewer 2 Report
The authors have successfully addressed all the issues raised by the reviewers. The manuscript can be published as is.